# Automatic (Tactile) Map Generation—A Systematic Literature Review

**Jakub Wabiński ***  **and Albina Mościcka**

Faculty of Civil Engineering and Geodesy, Military University of Technology, 00-908 Warsaw, Poland
* Correspondence: Jakub.wabinski@wat.edu.pl; Tel.: +48-601-533-573

**Abstract:** This paper presents a systematic literature review that reflects the current state of research in the field of algorithms and models for map generalization, the existing solutions for automatic (tactile) map generation, as well as good practices for designing spatial databases for the purposes of automatic map development. A total number of over 500 primary studies were screened in order to identify the most relevant research on automatic (tactile) map generation from the last decade. The reviewed papers revealed many existing solutions in the field of automatic map production, as well as algorithms (e.g., Douglas–Peucker, Visvalingam–Whyatt) and models (e.g., GAEL, CartACom) for data generalization that might be used to transform traditional spatial data into the haptic form, suitable for blind and visually impaired people. However, it turns out that a comprehensive solution for automatic tactile map generation does not exist.

**Keywords:** generalization; algorithm; model; map automation; tactile maps; maps for blind and visually impaired

## 1. Introduction

We are living in the Information Age—searching, browsing, downloading and archiving data is almost unlimited. This is owed to wide public access to the Internet. Besides, people simply want to know more about their environment and the world that they are living in. Much data has spatial reference and the best way to present that kind of data are maps. They have been used for thousands of years for multiple purposes, such as spatial planning, geology, forestry, and navigation [1].

Today, mainly digital maps are used. Such maps can be edited easily, and it is possible to generate them automatically or semi-automatically. In order to do that, it is necessary to use appropriate generalization algorithms that will allow developing a readable digital map and a smooth transition between scales. The generalization process is not linear and its results cannot be predicted only based on initial data and a set of rules or constraints. The final outcome is a consequence of numerous dynamic variables such as data richness, level of formalization and fuzziness of specifications [2]. Spatial data generalization is even more complicated in the case of tactile maps which are read using the sense of touch or, to a limited extent, using eyes [3].

The overall goal of this work is to provide an objective summary of the current state of research concerning automated map generation in general, but with particular emphasis on tactile maps. To the best of our knowledge, no significant literature reviews on this topic exist. However, as we will demonstrate in this article, there are numerous studies that deal with automatic map generation and many of them focus precisely on tactile maps. The reviewed articles allow us to draw certain conclusions regarding the potential of automatic (tactile) map generation; specifically the problems that have already been solved and the nature of the remaining challenges.

## 1.1. Background of Tactile Maps and Automatic Map Generation

Most people read maps using the sense of sight, which is the most natural way to perceive them. Unfortunately, disabled people who perceive the world with different senses cannot use them, so they cannot take full advantage of the Information Age. This group includes the blind and visually impaired. According to the World Health Organization 253 million people in the world are visually impaired, of which 36 million are blind and 217 million have moderate to severe vision impairment [4]. This is why we need to present spatial data in a form that is suitable for these people. One way of achieving this goal is to produce tactile maps. The main difference is that a sighted map user can cover the whole map sheet at once with his or her eyes, while a person with visual impairments has to read the tactile map fragment by fragment and must build up an image of the whole map in his or her memory [5]. However, apart from the differences resulting from the sense used to perceive the map, there are other characteristics of blind and visually impaired people that designers have to take into consideration. For example, blind and visually impaired people who suffer from diabetes have decreased tactile sensitivity and thus require different materials than those with high tactile acuity [6]. Simple conversion of the visual image into tangible form, regardless of the methodology used, would most probably lead to meaningless output [7]. Thankfully, along with the growing social awareness of the problems of disabled people, an increasing number of materials and solutions making their lives easier are appearing.

The first tactile maps were developed in the 18th century [8]. They were handcrafted for personal needs. Today tactile maps are usually printed with use of specific techniques that allow mass production, but the process of their creation is still expensive and time-consuming. The techniques that are currently used to produce these maps are suitable for production in large quantities. Thus, printing single copies of unique tactile maps is usually out of reach of those interested. Besides, developing a tactile map requires a team of specialists: tactile cartographers, tactile graphic designers, relief printing specialists and, when a map is designed to be used in schools, teachers and their pupils to validate its utility.

The Braille Authority of North America and Canadian Braille Authority [9] presented an overview of methods that are currently used to produce tactile maps. One of these methods involves embossers (e.g., Tiger, ViewPlus) to create tactile graphics from digital files. Embossed braille graphics can be produced easily and do not require physical masters, but at the same time, there is little variation in height between specific symbols and the number of possible textures that may be obtained is limited. Another production method is microcapsule graphics. They are printed on special microcapsule paper that are then extruded by heating its surface by a device called "enhancer." This method does not require physical master and graphics can be altered or duplicated. However, this technique requires many trials in order to achieve the desired results (e.g., appropriate extrusion heights). Besides, the microcapsule paper is rather expensive and can be damaged easily. Vacuum-formed graphics require a hard copy master. This method is preferred by many tactile map users due to its quality, but the production process is costly and time-consuming [10]. The aforementioned methods are well-established in the tactile map production industry but new solutions keep emerging, such as 3D printing [11], which is suitable for producing single copy maps at relatively low cost.

A new trend in tactile map production involves audio feedback. The ATMAPS Consortium, which consists of several European universities, public institutions, and a private company, is responsible for the project, whose main goal was to specify the audio, tactile and audio-tactile symbols that could be used in audio-tactile maps. Its results are described in a number of reports that can be found on the project's website [12]. Not only did the Consortium define standards for symbolization and map composition for five types of audio-tactile maps, but it also prepared an atlas of Europe that consists of 34, ready-to-print tactile maps (AT-atlas).

Apart from "static" techniques, there are a number of refreshable touch display technologies. They can be modified on-the-fly and often provide audio, vibration or haptic feedback [13]. These interactive solutions involve finger tracking systems or use touch-enabled surfaces [14]. A nice overview of refreshable

touch displays was presented by O'Modhrain et al. [7]. However, all these methods might benefit from improvements in production processes.

Standardization of tactile symbols might be a good first step to optimize the tactile map production process. Lobben and Lawrence [15] proposed a standardized set of tactile symbols designed for printing on microcapsule paper and made them available for free. Unfortunately, every printing technique requires different parameters of tactile symbols. This is why the development of a completely universal tactile symbols set might be impossible. Besides, tactile maps often require a decent degree of map content generalization and proper placement of symbols and labels. An average human without any visual impairment, under normal lighting conditions and at a viewing distance of 50 cm, can distinguish two points or lines as separate if they are at least 0.15 mm apart from each other [16]. To distinguish two points as separate with use of touch, they have to be at least 2.4 mm apart from each other [17,18]. Cartographic signs commonly used in traditional cartography are usually too small or too complicated to be read correctly using the sense of touch or a damaged sense of sight, even after raising them to a spatial form. This makes tactile maps less detailed and thus requires printing in larger formats [19].

However, the format of a tactile map is determined by the maximum reach of the user's arms. According to the survey results [20] the maps produced in the form of single sheet are more readable than those consisting of several parts. The aforementioned survey also deals with the topic of the maximum number of tactile signs that can be used on tactile maps. Map producers use 10 to 15 different signs on one map, out of which 6–7 are point signs, 3–4 are line signs and further 3–4 are surface signs (textures). When we compare these numbers with standards for classic map generation, the level of generalization required for tactile maps generation becomes even more evident. To facilitate map content distinction, tactile symbols should vary in height. According to widely-accepted standards [19], surface signs should be the lowest (0.5–1.0 mm), line signs moderate (1 mm), with point signs being the highest (1.5 mm). Braille characters have to be standardized. The most popular standard in Europe is the Marburger Medium Parameter, where the height of a Braille dot should fall between 0.5 and 0.8 mm.

Despite all the requirements mentioned above, a tactile cartographer has to bear in mind that the legibility of a tactile map is of the highest priority. Thus, tactile maps should meet the requirements related to tactile graphics design in the first place. The adherence to mathematical and cartographic principles is less important. An important step in the tactile maps production process is the end-user evaluation. The user feedback will vary depending on the test group characteristics and preferences, but it may prove invaluable in the process of proper tactile maps design.

On the other hand, access to spatial data has never been easier. Geoportals, spatial and statistical data, and software dedicated to map production—all these can be found on the Internet. However, cartographic skills are essential to produce correct maps manually, in contrast to automatic map generation. This process is also called "on-demand mapping" and described as automatic derivation of maps tailored to requirements expressed by users [21]. As stated by Armstrong [22], in the traditional map production process, the cartographer serves as the active agent. Even if a map is created automatically out of the set of spatial data, the cartographer has to iteratively modify it in order to achieve specific map user requirements. By having the data appropriately generalized in the first place, the automation of map production would be easier.

According to [23], "map generalization is a process of effective portraying changing levels of detail among geographic phenomena in order to reveal their various properties." Thus, cartographic generalization is used for transforming the original spatial dataset into maps of smaller scale. This involves changing the representation of spatial features and their placement on a map. Manual generalization is time-consuming and dependent on cognitive and technical skills as well as subjectivity of cartographer. The automation of this process speeds up map production and allows keeping all the maps up to date [24].

Automatic map generation is not a new research field [25–28]. However, it still requires a systematic review to provide a comprehensive summary of currently existing literature in this field. This particular review was prepared to obtain insight into the current state of research of automatic (tactile) map generation: methods, tools, and input data sources as well as the existing solutions. The results might be interesting not only for practitioners of cartography, but also for everyone else willing to involve maps in their daily work for spatial data representation. Today, due to democratization of cartography, everyone is a cartographer and can make his or her own maps [29]. Researchers, on the other hand, might learn about the current state of technology and find inspiration for future research.

### 1.2. Existing Literature Reviews and Motivation

We conducted a non-systematic keyword search in the libraries of selected journals, with the aim to find existing systematic literature reviews regarding automatic (tactile) map generation. The term searched was "systematic literature review" in a list of 30 journals.

This methodology allowed us to identify 20 literature reviews. Most of them are literature surveys and basic non-systematic reviews. However, based on their screening we decided to distinguish three systematic literature reviews. They are related to GIScience, cartography, remote sensing and climate change [30–32]. These reviews were used as sources of good practice examples for this particular systematic literature review. As none of them comply with the topic of automatic (tactile) map generation, they are insufficient to provide answers to general questions that were the motivation to take the presented research:

- What are the latest achievements and innovations in this field?
- What are the gaps in current research (if any)?
- Who currently conducts research on automatic (tactile) map generation?

A classic, non-systematic literature review would be insufficient to answer these questions, as it is usually of little scientific value and cannot be recreated. A systematic review synthesizes the existing works in a fair manner, as the review should be undertaken in accordance with a predefined search strategy, which ensures the completeness of the search [33]. A systematic review is an overview of primary studies, containing a clear statement of the objectives, materials, and methods and it is conducted according to explicit and reproducible methodology. Besides, as opposed to regular (journalistic) literature reviews, it includes results that might contradict the stated hypothesis [34].

## 2. Review Methodology

The main goals of this study are to review the possibilities of automatic map generation, especially tactile maps, as well as various concepts of data generalization. The review follows general systematic literature review guidelines [33]. Our review began with the development of a review protocol (Supplementary material). It specifies the background of the project, describes the inclusion/exclusion criteria, search strategy and methodology of data extraction, as well as the methods to synthesize and report results.

We started the review by defining clear and precise research questions, which enabled proper selection of primary studies [35]. The main task during automatic map generation is to use proper generalization algorithms and models (RQ1). Generalization might not be essential if there is good input data (RQ3). We were also interested in already existing solutions for automatic (tactile) map generation (RQ2). Taking the above into consideration, the following research questions were defined:

- RQ1: What are the generalization methods and models for automatic tactile/thematic/background map generation?
- RQ2: What are the existing systems and solutions allowing automatic (tactile) map generation?
- RQ3: How to design spatial databases for automatic map generation?

To determine whether primary studies are eligible for the systematic review, we applied the inclusion and exclusion criteria, defined in the Review Protocol.

We applied an iterative search strategy to determine the most suitable set of keywords to be used [33]. This also allowed us to minimize the search bias. The search strings described in the Review Protocol were used to browse seven electronic databases that included all the papers published before 31 June 2018. However, we tried to keep up to date with all the recent works on this subject. This is why we considered all the relevant articles manually gathered and suggested by scientific newsletters of Mendeley, ResearchGate and Tandfonline, even if they were published after the aforementioned date. This resulted in the identification of 646 digital sources (Table 1). All the selected studies were stored and managed within Mendeley Reference Management Software [36]. At the first stage, only title, keywords and abstract were analyzed. Two independent reviewers went through all of them and tagged them adequately using tags described in the Review Protocol.

**Table 1.** Electronic databases used during the review process (source: own study).

| SOURCE | URL | DATE OF SEARCH | SEARCH RESULTS |
|---|---|---|---|
| FREEFULLPDF | http://www.freefullpdf.com | 1 August 2018 | 38 |
| GOOGLE SCHOLAR | http://scholar.google.com | 31 July 2018 | 137 |
| IEEE LIBRARY | http://www.ieeexplore.ieee.org | 1 August 2018 | 85 |
| SCOPUS | https://www.scopus.com | 31 July 2018 | 176 |
| SPRINGER | https://link.springer.com | 1 August 2018 | 48 |
| WEB OF SCIENCE | http://www.webofknowledge.com | 31 July 2018 | 75 |
| WILLEY ONLINE LIBRARY | http://onlinelibrary.wiley.com | 1 August 2018 | 31 |
| OIN WAT * | n/a | 13 August 2018 | 29 |
| SCIENTIFIC NEWSLETTERS | n/a | n/a | 51 |

* Scientific Information Center of Military University of Technology in Warsaw.

Primary studies included by both reviewers were moved to quality assessment. Every inconsistency between the reviewers' choices was discussed. Thanks to these steps a set of 45 primary studies was selected for further analysis. An online spreadsheet that contains study quality assessment tables and data collection forms was developed for each of the primary studies (cf. Review Protocol). Papers that did not meet the criteria of minimum points for quality assessment were excluded. The same applies to papers that turned out to be completely irrelevant after full-paper screening, which was true for 11 papers.

After identifying the appropriate documents in online libraries, an iterative backward reference search was conducted in order to identify additional primary studies related to the topic of this review [35]. This resulted in 33 primary studies that went through full-paper screening. Only 11 of them complied with the inclusion criteria. Finally, 55 primary studies that show relevance to the research questions stated were selected for the review (Figure 1).

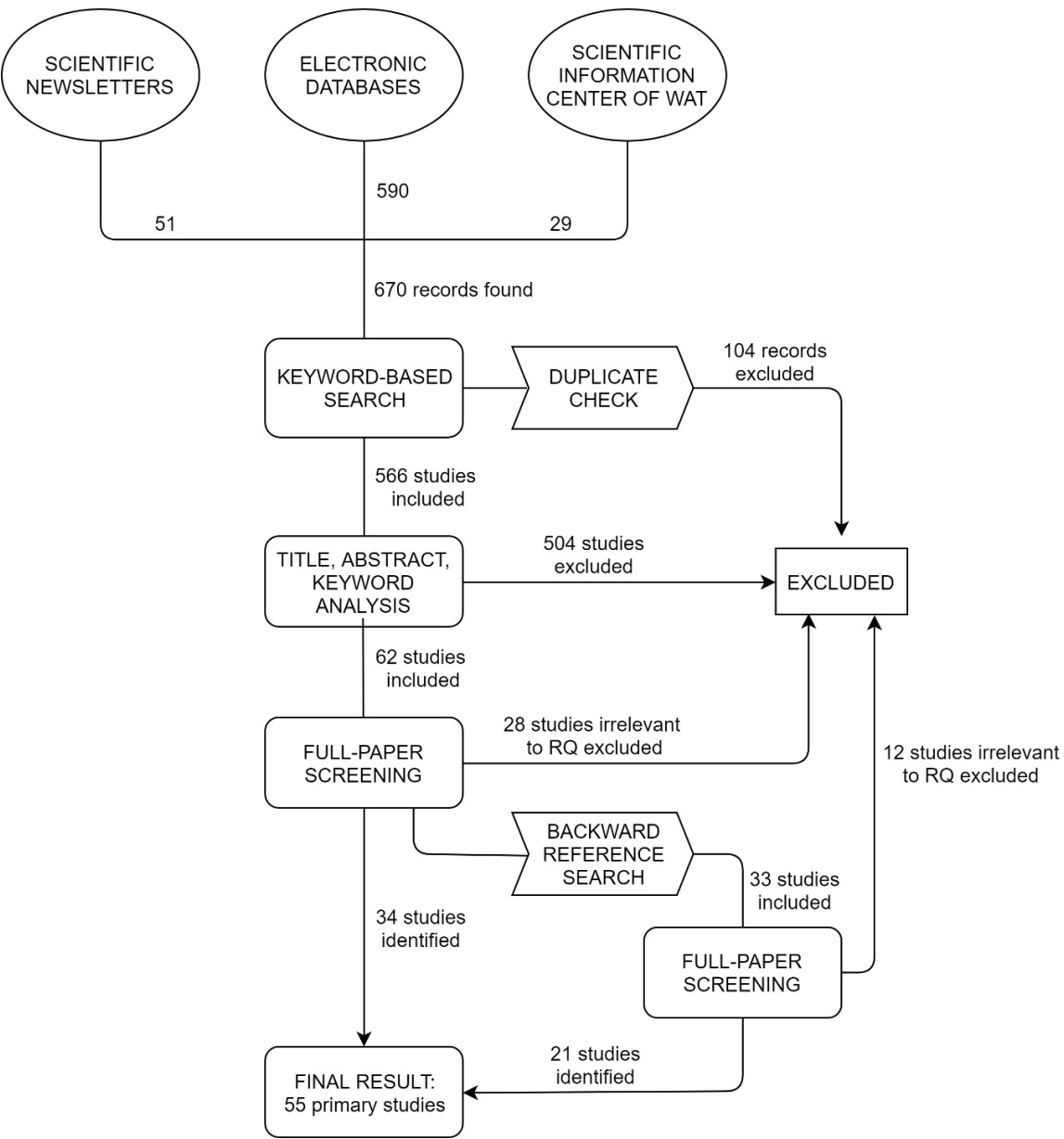

**Figure 1.** The course of the selection process together with the number of included and excluded papers in each step (source: own study).

## 3. Review Results

After a detailed analysis of the identified primary studies we can agree with the statements that appear in most of the articles, pointing out that automatic generalization processes are important in terms of automatic generation of both traditional and tactile maps [37–39]. It can also be noted that automatization of cartographic processes, which seemed impossible a few years ago, is now being successfully implemented in many countries.

The analysis of the year of publication of all identified papers did not reveal a clear trend (Figure 2). This applies to both the initial set of 566 identified works and the final set of 55 primary studies chosen for qualitative evaluation. Publications from 2019 were not covered in the systematic search and thus only five papers from this year were considered in this review.

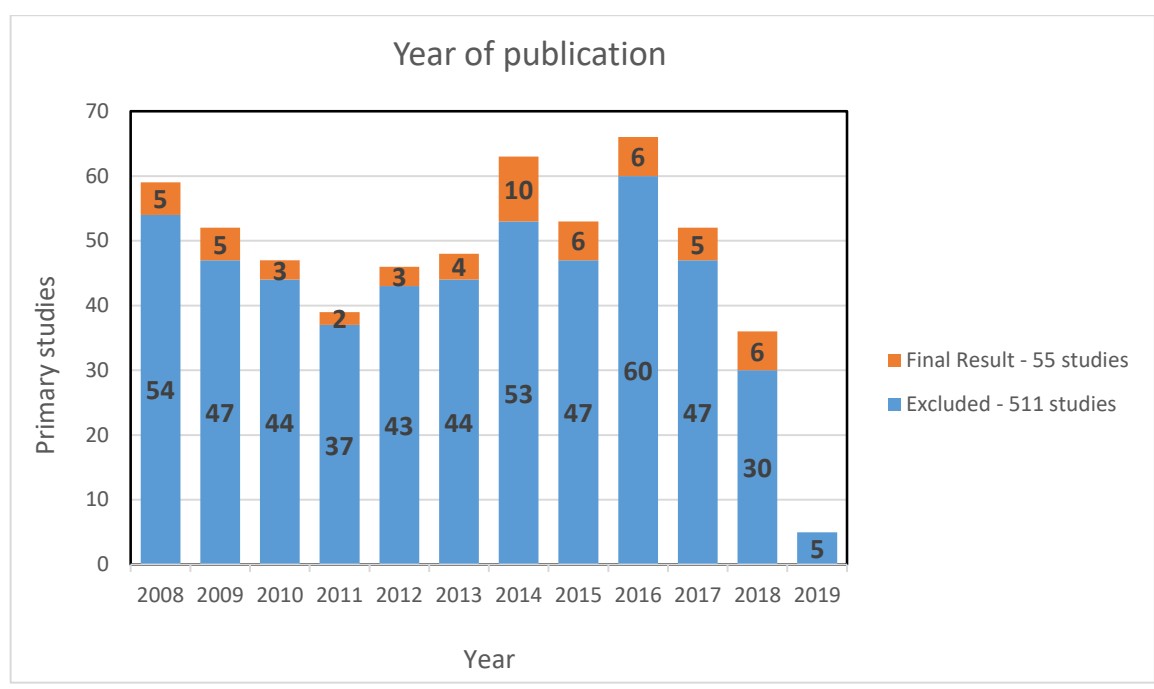

**Figure 2.** Relation between publish year and number of papers (source: own study).

During full-paper screening the tags were used to determine which of the stated research questions were answered by particular primary studies (cf. Review Protocol). More than half of the selected primary studies provided answers to RQ1, while 40% of them presented the existing systems and solutions in the field of automatic (tactile) map generation, providing answers to RQ2. Around one-third of papers presented good practices regarding spatial database design for automatic map generation, which was the answer to RQ3 (Figure 3).

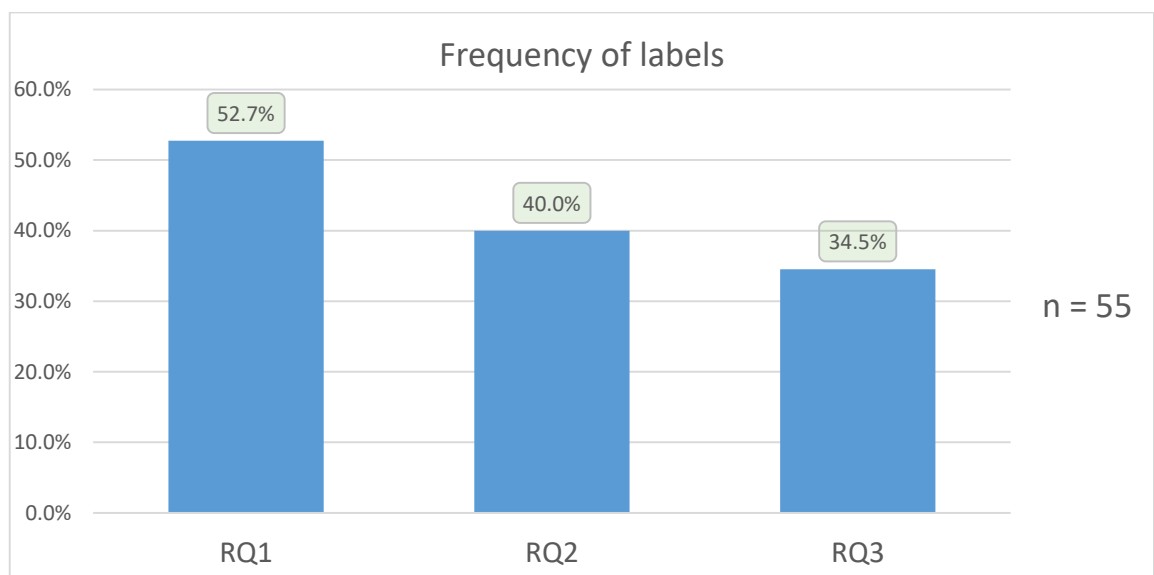

**Figure 3.** Number of primary studies providing answers to particular research questions (n—selected primary studies) (source: own study).

The associations between specific terms appearing most frequently within 55 selected primary studies were investigated, using weighted network visualization (Figure 4). This analysis was prepared in VOSviewer 1.6.9 using fractionalization method for normalizing the strength of the links between

items [40]. The bigger the label, the higher the weight of certain terms. The colors are determined by the cluster to which the term belongs, while lines represent links: the closer two terms appear, the stronger correlation between them exists. There are four clusters with the term "map" appearing most frequently across primary studies. The term "generalisation" appears very close to both "model" and "algorithm." "Map generalization" of "topographic data" is closely related to "scale" transitions, while the "tactile map" is correlated with people (users). Different spellings of the same words were not unified—this is why "generalization" and "generalisation" coexist in the figure.

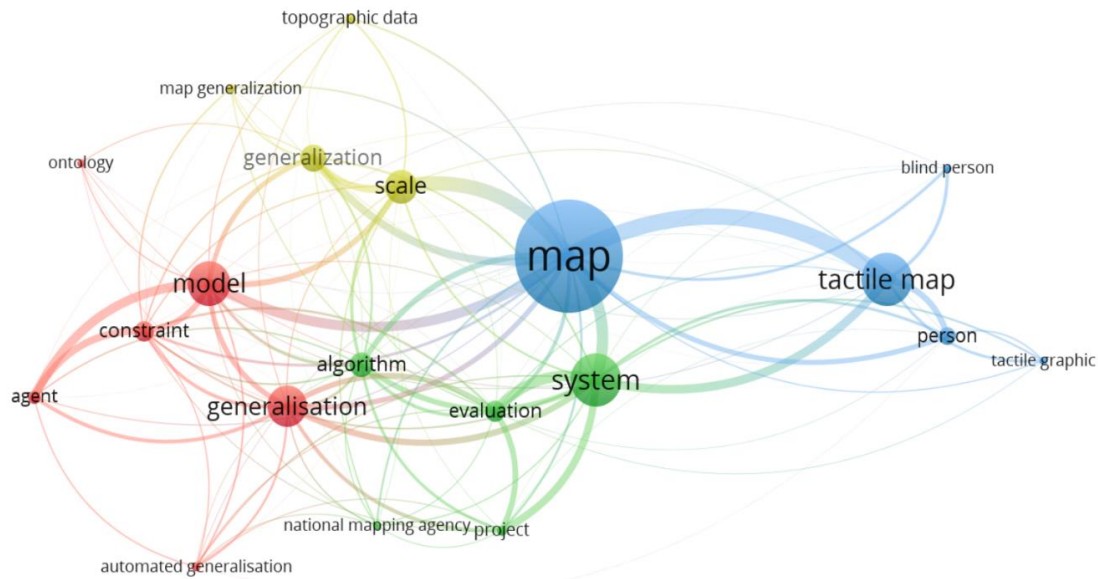

**Figure 4.** Weighted network visualization of associations between terms within identified primary studies (source: own study).

### 3.1. Research Question 1—What are the Generalization Methods and Models for Automatic Tactile/Thematic/Background Map Generation?

"Generalization methods and models for automatic map generation" was the most popular topic among the identified papers. Table 2 summarizes information regarding the identified generalization algorithms and models. It also provides a brief description of these studies. While preparing this section we were considering all appearances of the interesting features (algorithms, models, and operators) in the text. Not all tagged studies were included in the table.

We managed to extract many algorithms that can be used for the generalization of spatial data. *Douglas–Peucker* along with *Visvalingam–Whyatt* are well-known algorithms for simplifying line objects that appeared frequently across the studies [29,41]. The first one focuses on choosing line vertices to keep in the generalized version (bases on linear offset of each vertex), while the latter selects the ones to be deleted (uses area of displacement of each vertex). As a result, *Visvalingam–Whyatt* algorithm produces less angular results. The *elastic beams algorithm* for managing overlapping symbols treats linear features as elastic bars that bend if conflicts are detected [42]. *Skeletonization* that approximates medial axis for transformation of polygons into polylines [43], and *least-squares adjustment* is used for generalization in constraint-based approach where residuals are evenly distributed [44]. Other authors proposed modifications of existing solutions such as the *Douglas–Peucker–Peschier* algorithm [45], *combined stroke-mesh algorithm* for automated road selection during map scale transformations [46] or *collaborative displacement method* that combines aggregation, elimination and constrained reshape for building generalization in urban area maps [47]. In the given set of primary studies, we also identified proprietary solutions such as an algorithm for generation of abbreviated street names that was implemented because the names of features were too long to be placed on tactile map (written in Braille) [48]. A long list of algorithms for the optimization of label placement, such as the algorithm for

labelling islands by placing words outside their polygonal extent, as well as those for the simplification and smoothening of spatial features is presented by Reimer [49]. Takagi and Chen [50] described two additional algorithms that are useful for automatic map generation: *Hilditch's thinning algorithm* that obtains skeletons of scanned objects, which are then used for automated detection of hand-drawn features, and *Mamdani's fuzzy interference* used in the proposed system to design classification methods of the detected features.

The automatic generalization processes require a lot of computational power. The two most popular approaches for map generalization process distribution are *regular partitioning* and *geographical partitioning*. They are discussed in Berli et al. [51]. The first one is quicker, but it takes no contextual information into account, while the latter is less efficient in terms of running time, but results integrate the geographical context and in effect, is more accurate. The generalization models described in next paragraphs usually use one of these approaches.

Different generalization models are proposed across the reviewed literature. Some of them are purely conceptual, e.g., the *Pseudo-Physical Model* [52], while others are fully functional and tested solutions with certain applications: *Agent-based*, *GAEL* and *CartACom*. In the *Agent-based* model, each object of spatial database is considered an agent. This means that every entity of the database operates autonomously (without human intervention) and tries to achieve the desired goal using its capabilities—in this case to generalize itself in order to satisfy its cartographic constraints [53]. There are two levels of agents. A "micro" agent is a single geographic object (e.g., building), while a "meso" (macro) agent is a composition of "micro" or "meso" agents, as for generalization purposes they have to be considered together (e.g., a set of buildings in a neighborhood). This model is best used in dense [54] and well-structured urban areas [39]. On the other hand, the *GAEL* model (*Generalisation based on Agents and Elasticity*) is an extension of the *Agent-based* model for management of background themes such as relief. They differ from foreground themes because they are defined everywhere in the space. There are two types of cartographic constraints in *GAEL*: external, which cause the deformation, and internal, representing the shape preservation. The balance between them is required for successful generalization [55]. *GAEL* is often used for terrain models such as relief or land cover [54]. The *CartACom* model relies on communicating agents and was designed to handle unstructured geographic spaces with no clear borders between groups of objects [56]. Spatial relations between objects are of high importance in this model [57]. The model introduces relational constraints managing relations between two agents. There are three types of them: legibility constraints, constraints of preservation and of geographic coherence [58]. This model is best used for low density [54], heterogeneous rural areas [39]. In Touya and Duchêne [59] the authors argue that it is impossible to solve the generalization problem with a single model. The authors suggest that instead of constantly developing new solutions, one should benefit from the existing models and make them collaborate. This might be done by using the proposed framework—*Collaborative Generalisation*. In this approach specific map parts are treated by certain models that work best in particular cases.

A real issue in multi-scale mapping is the fact that there are two big gaps when changing scales: in terms of content and its representation [60]. This results in orientation problems while zooming in and out. The Authors present a new research project, whose goal is to reduce these problems by adding more intermediate levels (scales) to the already existing multi-scale image pyramid. More or less at the same time, in [61] a vario-scale structure for spatial data allowing zooming in and out smoothly was proposed. The Authors discuss how *tGAP* structure represented by 3D space-scale cube can facilitate continuous generalization.

During the review we counted the appearances of specific generalization operators in studies, using data extraction forms. It turned out that there was a big inconsistency in their naming—the review revealed 38 unique names of generalization operators across all the studies. This is because they come from different taxonomies and, as a result, they can bear various names while meaning the same. On the contrary, the same term may refer to different operators. There are three known operator taxonomies [28,62,63]. These taxonomies use natural language and if one would like to use them for

automatic map generation, it would be necessary to prepare formal and unequivocal descriptions of these operators [64]. Figure 5 presents the operators appearing in the identified primary studies normalized to the division proposed by Foerster et al. [63]. The most popular operator is *enhancement* (17% of appearances). It is used to emphasize the message carried by a spatial object in many ways (e.g., enlargement). Two other frequent operators are *displacement* (13%) usually used to maintain sufficient distances between map objects, and *simplification* (12%) for reducing the complexity of objects. *Reclassification* is rarely used (2%), which is understandable in view of the fact that most of the papers were describing generalization within National Mapping Agencies (NMAs), where reclassification of specific objects is undesirable.

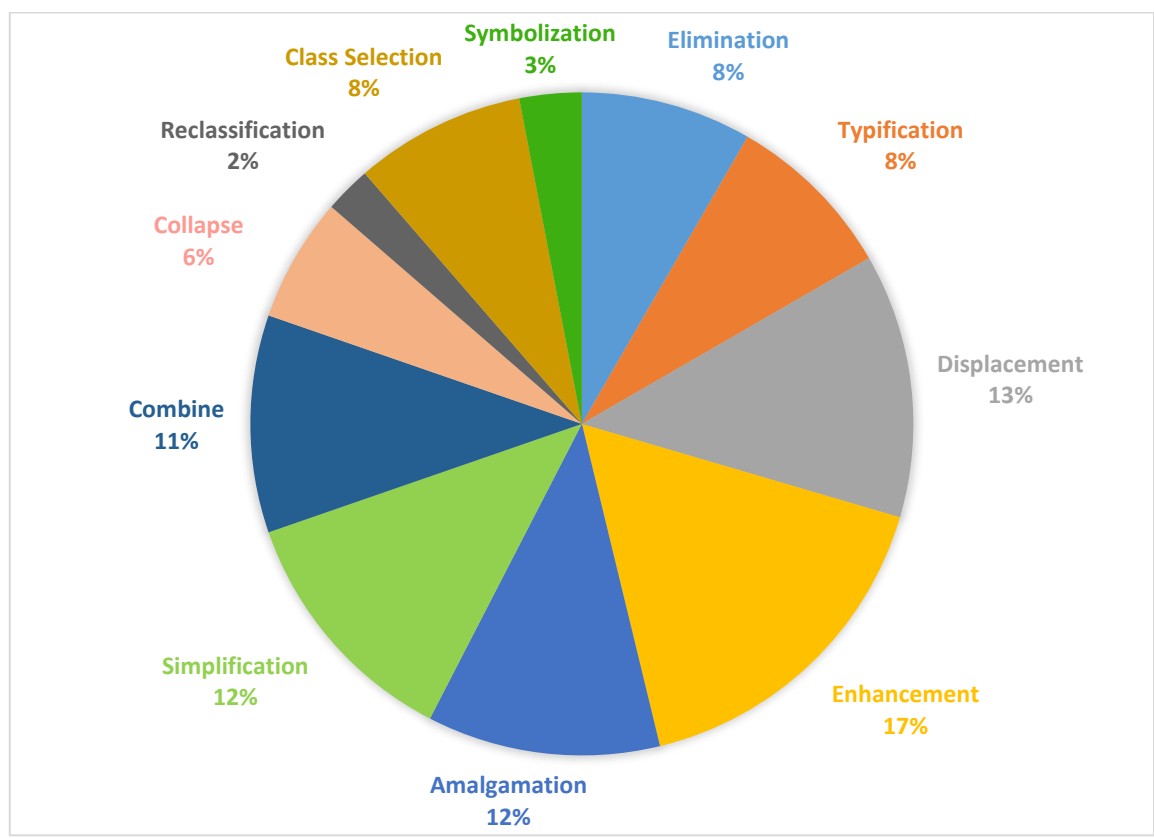

**Figure 5.** Number of appearances of generalization operators across primary studies (source: own study).

**Table 2.** Identified generalization algorithms and models (source: own work).

| Article | Generalization Algorithms | Generalization Models | Evaluation | Generalization Approach Overview |
|---|---|---|---|---|
| [37] | n/a | *General multi-scale conceptual model* | Tested on a sequence of maps of the Hunan lake region in China. | The authors propose multi-scale generalization operators. These are instructions for Agent's auto-generalization. |
| [52] | "Forces" control and determine each object's behavior (e.g., *clustering*, *reshaping*) | *Pseudo-physical model* (electric field theory) | Tested on a set of polyline and polygon objects in an urban area of Haifa in Israel. | The "power" of each object in any map is computed. It produces "forces" that act on objects and control them according to cartographic constraints. |
| [54] | n/a | *Multi-Agent-System CartACom GAEL Agent-based* | The approach is presented on sample bathymetric data in dedicated software. | Agents access cartographic knowledge stored in the ontology. The agent prepares different generalization plans regarding its environment. |
| [65] | The paper mentions NMAs that developed their own generalization algorithms, but it does not specify them. | n/a | n/a | Generalization operators used in NMAs in Europe. Which operators work best with certain feature types on maps, taking into account transitions between scales? |
| [60] | *Morphing of lines, continuous deformation of polygons, continuous vario-scale generalization* | *tGAP Space-Scale Cube (SSC) structure* | n/a | Providing intermediate levels to the multi-scale pyramid might allow smooth zooming and solve the problem of gaps between scales in existing map services. |
| [43] | *Skeletonization, morphological filtering, majority filter, simulated annealing, SVM, neural networks, fuzzy logic, random seed points* | n/a | Sample dataset of the Valencia region in Spain. The automatic method produced different results than the target dataset, but the results are promising. | Automatic generalization of land cover no-gaps polygon data from 1:1k to 1:25k scale. The methodology—several steps of aggregation of various feature types. |
| [66] | *Least squares adjustment, energy minimization, simulated annealing* | *Agent-based* | Tested on 99 sample urban blocks of Swisstopo data. Which of the eight tested operators were most frequently executed? | How to optimize a sequence of multiple generalization operators being applied to an entire set of map features? Eight operators in a sequence of maximum 20 steps were used. |
| [39] | *Least squares algorithm, Douglas Pecker, Displacement algorithm (CartAGen)* | *Agent-based CartACom GAEL* | Simple 3D printed map was created manually, followed by an attempt to obtain a similar map with use of automatic operations. The map was evaluated by experts. | Main research issues regarding models for automatic tactile map generation were identified. The authors concluded: no examples of automatic generalization, schematization and labelling in automatic tactile cartography. |
| [49] | A number of search algorithms are mentioned and others: *Rotating calipers, Visvalingam–Whyatt, Douglas–Peucker, Imai–Iri, Reimer–Meulemans* | *Sliding label, fixed position label models* Author's multi-criteria model for point feature labelling | Author tests his algorithm for labelling different types of features and then compares the results with manually generated labels. | Analysis of cartographic design principles for automated map generation. It focuses on map features labelling and proposes a model for this task that follows defined requirements and constraints. |
| [59] | *Simulated annealing, elastic beams, least squares algorithm* | *Agent-based GAEL Haunert* | Tested on very diverse sample data in France, coming from the "Official Publication N° 58 of Euro-SDR project." | The authors propose "Collaborative Generalisation" (CG). Instead of creating models from scratch; they would like to see the existing models work together. |
| [67] | *Elastic beams* | *Agent-based CartACom GAEL* | n/a | Automated generalization of vector data is based on synergy between three existing multi-agent generalization models: AGENT, CartACom, GAEL. |

**Table 2.** *Cont.*

| Article | Generalization Algorithms | Generalization Models | Evaluation | Generalization Approach Overview |
|---|---|---|---|---|
| [41] | *Douglas–Peucker, Visvalingam–Whyatt* | *Constrained tGAP* | Comparison of 1:10k scale data from constrained tGAP and existing 1:10k database. A number of issues were identified (to be fixed). | Constrained tGAP—model for deriving intermediate scales while having the same dataset available in two scales. What makes it different from tGAP standard is the way particular feature types are weighted. |
| [61] | *Douglas–Peucker* | *tGAP Space-Scale-Cube* | Three case studies presented varying in scale and type of data used. They are described in detail. | New approach to encode tGAP structure into SSC. This solution may support true smooth zoom generalization. However, there are still some open research questions. |
| [64] | n/a | n/a | Model based on case—visualization of road accident data. An attempt to formalize generalization (ontology). The results are promising. | Ontological modelling to articulate the knowledge used in automatic cartographic design. The authors try to create the ontology of generalization, which produces a map. |
| [45] | *Douglas–Peucker, Douglas–Peucker–Peschier, Visvalingam–Whyatt* | *Generalization Expert System* (GES) | Simplification of selected features from 1:250k to 1:500k map of Canberra, Australia. | Semi-automatic spatial data mining and generalization system for polygon and polyline data. Rule-based generalization expert system interfaced with ArcGIS. |
| [29] | Douglas–Peucker, *Visvalingham–Whyatt, Gaussian line smoothing, Perkal's E-circle rolling, least-squares fitting, spline interpolation, Fourier* or *wavelet transformation, skeletonization, center of gravity* | *Constraint-based* | n/a | The authors seek an answer to the question: "can map generalization automatically produce maps at a range of scales with minimum human intervention?" It consists of a quick review of the existing generalization algorithms and operators and it discusses the potential usage and current research conducted within NMAs. |
| [57] | n/a | *CartACom Topological relations: 4-intersection, 9-intersection, Region Connection Calculus* | Three case studies are presented regarding spatial relations in various situations. The proposed model requires improvement in order to be useful for automatic processes. | The book chapter presents a model for spatial relations ontology. These relations can then be used in automatic processes such as generalization or on-demand mapping. They can be quantitative or predicate/binary. The authors also propose four types of relational constraints. |
| [68] | *Growing tide, rural building "squared" amalgamation, weighted effective area algorithm* | n/a | After releasing the alpha version of the product feedback was collected from users. The beta version includes their suggestions. | The authors present a new product developed mostly automatically—"OS VectorMap District." Some of the generalization algorithms are also mentioned here. |
| [46] | *Stroke-based,* mesh-based, *combined stoke-mesh, graph-theoretic, extended DBSCAN* | *Constraint-based (soft/hard constraints)* | Evaluated by professional cartographers. Only 5–10% of the objects would need to be corrected manually. | Algorithm proposed for automated road network selection (transformation from 1:10k to 1:50k. Although it was designed for Swisstopo, it should work for other NMAs. |
| [44] | *Least squares, Douglas–Peucker, Gaussian smoothing, polygon merging, skeletonization, stroke-based road selection* | *ScaleMaster Agent-based* | ScaleMaster 2.0 was tested on VMAP of Abéché region in Chad. Results show that this model can be used to automatically derive DCMs from a MRDB, using several generalization processes. | The article proposes an extension of the ScaleMaster model. The new version is a model that drives automatic generalization and is readable by a generalization system, while the original version only provided descriptions and left the work to the cartographer. |
| [47] | *Collaborative displacement method, snake algorithm, elastic beams,* | *n/a* | Two topographic data sets—urban building maps in the 1:5k and 1:25k scales. The results indicate that the proposed method is effective, but some limitations exist. | The proposed method combines aggregation, elimination and constrained reshape operators. Vector field-based displacement is adopted. If it fails, then the proposed method is used. |

SVM—Support Vector Machine; NMA—National Mapping Agency; GAEL—Generalisation based on Agents and Elasticity; tGAP—topological Generalized Area Partition; VMAP—Vector Smart Map; MRDB—Multi-Resolution Database; DCMs—Digital Cartographic Models.

A detailed study by Foerster et al. [65] presented a quantitative analysis of generalization operators as indicators of the current status of automatic map generalization at NMAs in Europe. In a set of figures, the authors show the importance of operators in relation to spatial feature types most frequently appearing on maps (e.g., buildings, relief), with respect to transitions between scales. Neun, Burghardt, and Weibel [66] in their work use three different search algorithms (*hill climbing, simulated annealing, genetic deep search*) to determine the best sequence of generalization operators applied to a specific set of spatial data—urban blocks. They were evaluated in terms of processing time and the amount of conflict-reduction.

The authors of identified primary studies used various software. Some authors mentioned their own dedicated software or system [54,69–71]. In most cases, however, researchers used commercial software and, if necessary, modified it to suit their needs. S Kazemi et al. [45] proposed a knowledge-based solution build in Java-Python that is interfaced with ESRI ArcGIS for automatic generalization of thematic data (polylines and polygons). Stoter et al. [2] in their report described tests performed in years 2007–2008, where project team members used unmodified versions of commercial generalization systems (*ArcGIS, Axpand, Change/Push/Typify, Radius Clarity*). The main goal was to show the possibilities and limitations of existing commercial generalization software and to determine whether these systems would match the generalization criteria of specific NMAs. They analyzed four test cases but none of them was fully generalized by the unmodified systems. The Authors suggested that customization of these systems may provide acceptable results in terms of automatic generalization [2]. This had to be true as a couple of years later first fully automatic solutions began to appear [38,48,68].

*3.2. Research Question 2—What Are the Existing Systems and Solutions Allowing Automatic (Tactile) Map Generation?*

The review revealed that numerous solutions regarding automatic generation of maps for the blind and visually impaired exist. It identified solutions that are usually designed for the needs of NMAs regarding automatic generation of topographic maps or to allow generation of orientation and navigation tactile maps. A brief summary of our review is presented in Table 3.

**Table 3.** Existing systems and solutions allowing automatic (tactile) map generation (source: own work).

| Article | Type of Maps | Operating Range | Tactile | Name | Study Overview |
|---|---|---|---|---|---|
| [50,71,72] | Orientation and navigation | Designed for Japan but would work everywhere (Range of OSM data) | Yes | Tactile Map Automated Creation System (TMACS) | Computer-aided platform for automatic translation of hand-drawn maps into tactile maps. In 2014 it was modified to handle OpenStreetMap data. |
| [73] | Topographic | Germany | No | ATKIS-Gen: Amtliches Topographisch-Kartographisches Informationssystem Generalisierung | Automatic generalization system using AGENT-Technology of 1Spatial. All the products are derived from basis DLM using model and cartographic generalization. |
| [48] | Orientation and navigation | City of Brno, Czech Republic | Yes | n/a | System used for partly automatic creation of a set of 1:2 500 orientation maps, ready for relief printing on microcapsule paper. Map sheets are combinable into larger areas. |
| [69] | Destination (navigation) | Wherever Bing Maps are available | No | Destination Maps under Map Apps (discontinued) | Fully automated system for creating destination maps that is based on principles used by mapmakers. The system simplifies selected roads, optimizes their position, scale and orientation, and adds geographic contextual information. |
| [74] | Tourist | Determined by the database content (e.g., 3D buildings) | No | n/a | Automatic generation of tourist maps based on an existing database, which is constantly updated (according to authors). |
| [75] | Topographic, Historical | Whole world (depends on map type selected) | No | Carte-a-la-carte | Existing system enabling customers to define a customized paper map (not free of charge). It is possible to include the title and a cover illustration (logo), using three kinds of maps. |
| [70] | Road atlas | Global | Yes | Mapy.cz | Conventional map underlays are adjusted so that they can be printed on microcapsule paper and used by blind people. |
| [76–78] | Orientation and navigation | Range of OSM data | Yes | Blindweb.org | Platform for automatic generation of tactile maps based on OSM. It allows creation of graphics for 3D printing (also audio-haptic overlays) and microcapsule paper. |
| [79] | Orientation and navigation | Range of OSM data | Yes | Tactilemaps.net | Complete end-to-end system that allows the blind and visually impaired to independently generate tactile maps. Users can either generate a 3D model or order a print. |
| [80,81] | Orientation and navigation | Range of OSM data | Yes | HaptOSM | The solution is a combination of specialized hardware and software based on OSM data that allows creating individual tactile maps almost entirely automatically. |
| [38,82] | Topographic | The Netherlands | No | n/a | Successful methodology for automatic derivation of 1:50k maps from 1:10k data. |
| [42] | Topographic | Catalonia, France, Germany, Switzerland, Great Britain, the U.S., the Netherlands, | No | n/a | Description of automated generalization carried out in seven chosen NMAs (maps, label placement). |
| [83] | Orientation and navigation | Range of MapQuest (not clearly specified) | Yes | n/a | System for automatic conversion of JPEG images into graphics that can be used in braille embossers and microcapsule paper. |
| [84] | Land cover | Germany | No | CLC-generator | Methodology of land-cover datasets automatic generalization from topographic data successfully implemented for conversion of Basis DLM into CORINE Land Cover. |

OSM—OpenStreetMap; DLM—Digital Landscape Mode.

In March 2013 a symposium on "Generalization within NMAs" was held in Barcelona. Duchêne et al. [42] prepared a review of the state-of-the-art within NMAs regarding automatic map generalization. Based on contributions from seven interviewed NMAs and conclusions drawn after the symposium we can assume that, as of the year 2014, the selected NMAs had made significant progress. It turned out that all considered NMAs had managed to renew their data models so that they took a form of structured spatial databases, characterized by consistency between particular levels of details. Eleven out of 12 NMAs had introduced either automatic or semi-automatic generalization processes. According to the report, examples of successful implementation of fully automatic generalization existed in Ordnance Survey Great Britain where they managed to derive 1:25k Digital Cartographic Model out of a mixed-scale Digital Landscape Model (DLM), at IGN France—1:25k Digital Cartographic Model (DCM) from a 1:10k DLM, and at Kadaster Netherlands—1:50k DCM from a 1:10k DLM. Such works facilitate the shortening of update cycles of spatial databases but are still insufficient as far as the needs of modern customers are concerned.

An interesting project in the field of automatic generalization is described by Thiemann and Sester [84]. The developed program called *CLC-Generator* allows automatic generation of Corine Land Cover data that is updated every six years from the high resolution German land-cover dataset with update rate of one year. Authors present an example workflow with generalization operators and their parameters specified.

We also identified certain solutions designed to produce thematic maps. Kopf et al. [69] presented an already implemented system for creating unique destination maps. After the user has defined the destination point, the system first simplifies the geometry of roads within a predefined range, then optimizes their position, scale and orientation in a non-uniform way, and finally adds contextual geographic information to facilitate orientation. A different solution was presented in Grabler et al. [74]. It allows for automatic generation of tourist maps. These maps highlight touristic points of interest, so that they are not only useful but also good-looking. Many commercial platforms for automatic map generation can be found online. The review revealed one that is managed by IGN France [75]. Users can order personalized maps (e.g., historical maps, or orthophotomaps) in either digital or paper form. However, this type of solution does not apply any generalization but simply downloads the necessary files from a dedicated database.

We found numerous solutions dedicated to blind and visually impaired users. Many of them are automatic systems that are based on *Open Street Map* (OSM) data. One of the examples is the *Tactile Map Automated Creation System* (TMACS). First, its authors developed a computer-aided platform that was able to recognize hand-drawn images of maps and translate them into digital tactile maps [50,71]. In 2014, the system was modified to allow automatic tactile map production of any given location in the world [72]. Based on the address entered by the user, the system generates an image that can then be printed using microcapsule paper and raised to a spatial (tactile) form. *HaptOSM* is another example of a complete system. It consists of software that uses data from OSM along with additional parameters and converts them into G-Code, and hardware: a dedicated CNC-Router that embosses map data on Braille paper of writing film [80,81]. A different system based on OSM data was described in Taylor et al. [79]. Their online platform [85] is adapted to be used by visually impaired people thanks to screen-reader compatibility. The platform has two interfaces. In simple interface users are only asked to provide an address and specify the size of the map. This is enough to generate a 3D model of roadways in the area. The advanced interface provides additional options for map customization: users can add additional features such as waterways and points of interest. Users can either order a 3D model that can be then 3D printed or directly order a physical print of their map. The system can also generate physical maps, using conductive filament. They can work as interactive overlays for touchscreens to provide dynamic touch interactivity with the maps. A more extensive system, also based on OSM data, was presented in Götzelmann and Eichler [76], Götzelmann and Pavkovic [77]. In this case not only the platform [86] can be read by screen readers but also an actual map image is generated based on user requirements. To generate a map, users have to specify an address that determines the central point of

the map and choose the map features to be represented on the map sorted in categories (e.g., Health, Accessibility), map zoom level and output technology. This system is capable of generating files for 3D printing (including audio-haptic overlays), printing on microcapsule paper or compatible with braille embossers. Further developments of this platform were discussed in [78]. The paper presented a solution for audio-tactile overlays working with usual smartphones or tablets that can be 3D printed using conductive materials. Special capacitive codes are used to identify particular tactile maps being placed on a touch screen and provide audio feedback to blind or visually impaired users. The system proposed by [83] is based on MapQuest and uses digital map images. It first detects texts and extracts them so that they are processed separately from graphics. Features identified in this way are then translated into a form appropriate for tactile printing and integrated in Support Vector Machine form that enables map exploration with use of a touchpad. The system performs these operations fully automatically by just providing an input file. Output files can be printed either on microcapsule paper or using braille embosser and augmented with audio descriptions.

Unfortunately, such collaborative mapping platforms as OSM are created by people who are often unexperienced "mapmakers." These data are characterized by strong heterogeneity [87], which results in inconsistencies and topological flaws that make them unreliable. Due to that, it would be best not to use such data for automatic tactile map generation. However, OSM offers impressive coverage and enables to generate maps of almost every place on Earth. Besides, due to lack of communication between NMAs, there are few examples of cooperation between countries to provide spatial data of high cartographic quality for automatic tactile map generation. However, some actually exist. Štampach and Mulíčková [48] presented a system developed by joint efforts of public institutions and academic environment in form of Python scripts for ESRI ArcGIS software that allowed them to generate a set of 1:2500 orientation and navigation maps semi-automatically. The grid of the generated map sheets covers the whole area of Brno City in Czech Republic. Individual tactile maps were prepared to be relief printed using microcapsule paper and can be combined to form larger areas. Each map follows the proposed scheme and complies with the requirements provided by the Support Centre for Students with Special Needs at Masaryk University. Another example of existing mapping solution in Czech Republic is the one described in Červenka et al. [70]. The paper described an enhancement that was prepared for the official mapping service Mapy.cz [88]. It adjusted traditional map underlays into a form that can be printed on microcapsule paper and used by blind and visually impaired people. Again, the platform was created thanks to cooperation of public and educational institutions, as well as commercial sector.

Some of the papers did not mention any specific solutions but presented planned research in the field of automatic tactile map generation. Ducasse, Macé, and Jouffrais [89] provided information about the *AcessiMap* research project, whose goal is to improve map accessibility for the visually impaired in France. The paper also mentioned other existing solutions in the field of tactile map generation such as *Touch the map!*. It enables interaction of blind users with tactile maps (e.g., retrieving names and distances using gestures). Some commercial solutions were also presented, including *HyperBraille* for creating dynamic pin-raised displays or *SpaceSence* and T*ouchOverMap*, that rely on touch sensitive surfaces without any overlays. Another project mentioned in the paper is the one that is often referred to as a pioneering project in the field of automatic tactile map generation—*Talking TMAP*. The project was initiated in 2003 and resulted in creating a web-based software tool for fast production of personalized tactile street maps of any location in the United States. They could then be downloaded and personally embossed [90].

Current trends and a summary of research status in on-demand tactile maps generation were gathered in a review article [39]. Apart from studies identified during the review, there are two other projects worth mentioning. Schwarzbach et al. in [91] proposed a solution that allows producing colorful 3D printed maps facilitating teaching geography to blind and visually impaired people. Using similar technology enables to generate 3D models from LIDAR data and orthophotomaps [92].

### 3.3. Research Question 3—How to Properly Design Spatial Databases for Automatic Map Generation?

The importance of creating unified geographical datasets (RQ3) for automatic (on-demand) tactile map generation was stated in Guillaume Touya et al. [39]. The solutions used in NMAs and described in the previous sections employed systems for automatic map generation that derived consecutive DLMs or DCMs from the main spatial database. The question is how to design such database? What requirements should it match? Most of the identified primary studies provided little information on this topic. This is probably because specifications of spatial databases for automatic map generation are either classified or described in formal documents rather than scientific publications. However, there are examples of important works in this field. An exhaustive research regarding vario-scale database structure was described in [93]. The main research question of this study was how to design a system dedicated for vario-scale mapping and present a working pipeline from the pre-processing steps to the final data visualization. Examples of base datasets for automatic map generalization were described in [65]. The authors mentioned Dutch *Multi-Scale Information Model Topography* (IMTOP), Danish multi-scale *GeoDB*, varying-scale *OS MasterMap* produced in Great Britain. In France, the *IGN BDTOPO* database is updated in six-months' cycles and is used to produce multi-scale vector data [75]. The authors pointed out that in many situations it is more important to have a database that is able to produce less-detailed up-to-date results, rather than very precise but out of date topographic maps. In many cases up-to-dateness may be of higher priority than cartographic rules [82]. Nevertheless, the resulting maps must comply with basic quality requirements. In Switzerland, the *Topographic Landscape Model 3D* (TLM3D) serves as the master database out of which topographic maps and other database products are derived [46]. In Poland, the basis for the national spatial data infrastructure is the 1:10k *Topographic Objects Database* (BDOT) that is used for semi-automatic derivation of the 1:250k *General Geographic Database (BDOO)*. Currently, full automation of this process is one of the main goals of Polish NMA [94]. However, as it was recently stated in [95], it is not possible to perform automatic generalization of BDOT10k, using out-of-the-box software. In the Netherlands, a decade ago the authorities wanted to develop a new large-scale topographic standard of a vario-scale database—*IMGeo*. One of the ideas proposed by Hofman [96] was to make the *IMGeo* model based on existing constraints from the *Top10NL* standard using constrained *tGAP* approach. The main focus of this work was on generalizing polygon features. The requirement for *tGAP* approach is that area partition should cover the whole map. That turned out to be a real problem as the *Top10NL* structure did not match this criterion [96]. Constrained *tGAP* approach can be used to derive intermediate scales of maps but with one limitation—a final scale dataset has to be known a priori [61].

In general, two different approaches are being used by NMAs regarding the way certain products are created out of the main DLM: "star" and "ladder." In the "star" approach every DLM or DCM is derived directly from the main database through generalization, while in the "ladder" approach the base DLM is generalized to form a lower resolution DLM or DCM, which is then generalized into an even coarser DLM or DCM [42]. According to the report by Cécile Duchêne et al. [42] the only NMA using the "star" approach is Ordnance Survey Great Britain. The rest of the NMAs use a mixed approach (e.g., [73]). Most of the NMAs (eight out of 12) decided to implement Multi-Resolution Database (MRDB) that allows maintaining links between different DLMs and DCMs.

The problem with these databases is that they are not interrelated. Globally available spatial data such as OSM are unreliable (cf. Section 3.2—RQ2), but they can be dynamically updated. Besides, some researchers suggest that solutions for on-demand generation of tactile maps should rely on the kind of spatial data that are free to use and publicly available [89].

An idea on how to use a certain database structure for storing planar spatial data was presented in Xiang, Huang, Shao and Wang [97]. The authors proposed their implementation of popular NoSQL MongoDB and evaluated it as a highly capable, rich query language choice to manage spatial data. Their main goal was to combine the strengths of both MongoDB and R-tree structure to support geodetic spatial data management and to provide queries with spatial predicates, while at the same time speeding up the query processing. This will influence the process of selecting appropriate objects

for map generalization. There is a close correlation between spatial data and the constraints that determine their generalization. Stoter et al. [98] argued that they should be as formal as possible to support the generalization process and its evaluation. The same applies to databases used for automatic derivation of maps. They have to be consistent, unified with other databases, and topologically correct.

An issue that has been tackled in Boedecker [99] is the simultaneous generalization and coordination of various feature types. In order to solve this problem properly, a generic set of cartographic relations that will be preserved during generalization has to be defined. Data enrichment is one of the ways to preserve them. This should make the generalization level of different thematic features well-balanced. These cartographic relations (namely: geometric, topological, statistical and semantic) support the coordination of different map themes during generalization. A similar issue was discussed in Jaara, Duchêne, and Ruas [100]. In the real world, reference and thematic data might come from various sources. A process called thematic data migration is essential to maintain data consistency, for example while merging thematic data from different sources. The authors presented a methodology based on "relation satisfaction measure" as a good way of evaluating thematic data migration. This issue might be avoided if one unified master database existed. Even so, great caution has to be taken while using such master database. Kazemi et al. [101] argued that, during generalization, an initial decision on how much initial data should be discarded from the master database results in a more varied final dataset than the choice of certain generalization operators or algorithms. This might be an obvious statement, but it clearly shows how important it is to design a spatial database properly.

## 4. Discussion

We can assume from full-text analysis that most of the evaluated existing solutions in the field of automatic (tactile) map generation (RQ2), were described in primary studies from years 2014–2018. This is especially true for online platforms for tactile map generation. We were happy to see that the number of systems for automatic map generation is growing constantly. At the same time, we are concerned about the fact that many of them seem to have been discontinued. We are unsure if this is because of lack of funds or because these solutions proved to be useless in the real world outside the lab. Another problem is that even though researchers can currently communicate freely and have almost unlimited access to knowledge, collaboration between them seems scarce. Solutions produced in different countries or even different research centers are not transferred and shared. This is also true for spatial databases used for automatic generation of various cartographic products, which is even more surprising as most of the solutions developed for the blind and visually impaired are based on worldwide available data from open mapping projects such as OSM. In general, more than half of all systems identified in the chosen set of primary studies were designed for blind and visually impaired people. However, none of them deals with thematic tactile maps used in education. Most of the identified systems are designed for generating orientation and navigation maps.

Automatic map generation is an optimization issue for many NMAs. Their aim is to transform high-scale spatial databases into lower-scale topographic maps using model and cartographic generalization. The most popular challenge for NMAs is to create intermediate levels of maps based on existing spatial databases. Besides, there is an on-going discussion on whether it is better to prioritize the up-to-dateness or the quality of these databases. In a modern, dynamic world people expect to have all the information immediately, but NMAs are institutions that should take extreme care regarding quality of the data they provide. Although this might be the result of search bias, it looks like most of the work in the investigated field is currently being carried out in Europe, the U.S. and Japan.

After attempting to answer RQ1 regarding the best algorithms and models for automatic map generalization, we can conclude that this question remains open. Despite the amount of time researchers devoted to it in recent years, there are still no holistic solutions in this field. It might be interesting to note that according to some experts one does not need new models but a methodology that will allow to combine them and thus realize their full potential [59]. A confirmation of this statement can be found while analyzing algorithms used for automatic map generalization. Most of the studies used

popular algorithms such as *Douglas–Peucker* or their modifications that have been known for years. Moreover, most of the identified models used a constraint-based approach. Perhaps we do not need something completely new to solve the evergreen issue of automatic generalization?

We believe that building a unified, verified and up-to-date spatial database is a good starting point. In times of Volunteered Geographic Information, when people provide their location by georeferencing photos or social media posts and cooperate in projects such as OSM to build global cartographic coverage, this "elusive dream" might become reality. Some examples of such master databases already exist (cf. Section 3.2), but this is not enough. Again, instead of building something completely new, as in the case of generalization models, one should consider adapting the existing solutions. The Open Street Map project has great potential, but it has to be formalized. In our opinion, some full-time data validators should be employed in order to ensure the right quality of the provided data.

We believe that this systematic review presents a brief overview of the current state of research in the field of automatic (tactile) map generation, and that it will provide new directions for research and development. To summarize, the review presents the current gaps in the research and issues identified during review:

1. Despite various existing generalization algorithms and models, a holistic solution that would be able to process the entire map at once does not exist;
2. There is no collaboration between researchers and NMAs dealing with this topic. Many of the working solutions were never implemented outside the country of origin;
3. The existing solutions for automatic tactile map generation are based on unmodified data. Spatial data has to be transformed into the form legible for the blind and visually impaired first;
4. There are no existing solutions for generating automatic thematic tactile map that could be used in education;
5. Despite the existence of the European Union INSPIRE Directive, the spatial databases of EU member countries are still not fully compatible.

The reviewed papers as well as the identified gaps form the basis for a research agenda regarding automatic tactile map generation. In our opinion it might be a good idea to transfer the verified and working solutions developed for automatic generation of traditional maps into the field of tactile cartography, which seems to be more reasonable than developing entirely new solutions from scratch. However, this requires developing generalization rules customized to the perceptual possibilities of the blind and using a different approach to map composition, having in mind the specific requirements of tactile maps design. As in traditional cartography, we should develop rules for updating such maps, as well as try to use already existing data, generalized to a level acceptable to blind people, in the development of new maps. It can reduce the time of developing maps for the blind and reduce their costs. While working on this, researchers should try to implement the principles of universal design. Currently a lot of effort is being put in the unification of various databases, such as in the case of European Union and thus, international cooperation between research centers and public institutions should be of high importance in future projects.

As a number of working solutions for orientation and mobility tactile maps exist, researchers should now focus on educational tactile maps. We think that so far there has been too little research in this field. This kind of map is necessary for the proper implementation of the curriculum for blind and visually impaired children. Future projects should take advantage of new production techniques, such as 3D printing, and implement the described ideas to make the maps more interactive (e.g., refreshable displays, audio feedback).

At this point we have to discuss the limitations of this review. First of all, we did not cover all the existing electronic databases but only selected ones. Analogue materials were not deeply investigated, either. Besides, the electronic databases differed from each other. They were based on various search engines that required different search queries. Nevertheless, we tried to modify them so that they would provide similar results.

This review is also prone to search bias as during keyword-based search we used those keywords that are influenced by our academic background. As a result, some studies in other disciplines might have remained undiscovered. The process of Title–Abstract–Keywords analysis is subjective and this is why this phase was conducted by two independent reviewers in order to minimize bias. To ensure that the quality of the chosen studies was high, we used quality assessment forms during full-paper screening. Studies that did not receive the required score were also excluded (504). In order to allow reproduction of this step in the future we tagged all the excluded studies in a manner described in the Review Protocol (Figure 6).

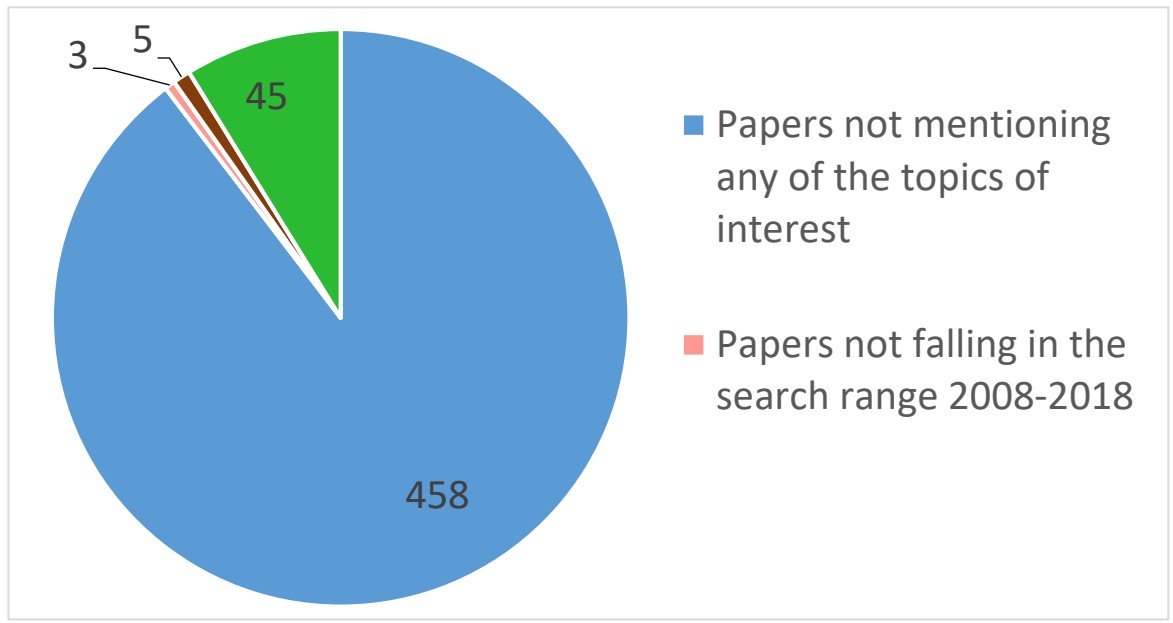

**Figure 6.** Reasons for the exclusion of certain papers from the review (source: own study).

The most popular reason for the exclusion of certain studies was the fact that they were unrelated to any of the research questions (458). This decision was made without full-paper screening and thus, it might be biased. Another limiting aspect was that some of the studies were written in languages that we were unfamiliar with (5). Some of the papers' metadata contained false information about the publication year. Those published before 2008 were excluded (3). We also have to mention that most of the studies' metadata was not machine-readable and, as a result, we had to conduct some of the presented analyses manually. The fact that we analyzed only the research from the last decade is another limitation. During backward reference search we also chose only studies from years 2008–2018. In fact, backward reference search was also subjective because it was based on Title–Abstract–Keywords analysis. Due to time limitations this process was conducted only on studies selected during full-paper screening (34). However, there might be some valuable research papers among the citations of the initially identified studies (566).

## 5. Conclusions

This article presents a systematic literature review on the research status regarding automatic (tactile) map generation, with a focus on generalization algorithms and models, existing solutions and methodology of spatial database design to serve this purpose. By using transparent workflow described in the Review Protocol, we aimed to create a review that is replicable and reproducible. To provide an exhaustive data collection we analyzed results from many heterogeneous and interdisciplinary digital libraries. Our intention was to make this review free of bias by applying different quantitative and qualitative techniques, but obviously this was impossible. All the primary studies that we analyzed are available for researchers to investigate but unfortunately often not for free and thus, out of reach

of laypersons. This is where our review might turn out to be useful. Fortunately, currently there is a strong movement to make all the research accessible to the general public and hopefully one day science will be open to everyone.

We managed to answer our research questions during the review. The overview of the existing methods might be a motivation for researchers from across the globe to cooperate. The review also identified the existing research gaps and proved that there is a strong need for new research contributions in this field. This might be an inspiration for many researchers. In conclusion, much has already been said but there is still a lot to do.

**Supplementary Materials:** The following are available online at http://www.mdpi.com/2220-9964/8/7/293/s1, Document S1: Automatic (tactile) map generation—Review Protocol.

**Author Contributions:** Jakub Wabiński designed the study and identified digital sources; Jakub Wabiński and Albina Mościcka performed the primary studies screening and quality assessment; Jakub Wabiński wrote the paper and Albina Mościcka made corrections to the text.

**Funding:** This research was funded by the Military University of Technology in Warsaw, Poland under the Young Scientist's Program grant titled "Modern methods of production in cartography and automation of the process of generating thematic maps," grant number RMN 850/2018, realized in the years 2018–2019.

**Acknowledgments:** Heartfelt thanks go to Translation Office Lingua Line as well as English translator Antonina Kozłowska for language correction of the text.

**Conflicts of Interest:** We report no conflict of interest. The funding sponsors had no role in the design of the study; in the collection, analyses, or interpretation of data; in the writing of the manuscript, and in the decision to publish the results.

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
