# Peer review of "Automatic (Tactile) Map Generation—A Systematic Literature Review"

_ijgi, doi:10.3390/ijgi8070293_

Round 1

Reviewer 1 Report

From point of view to present completed overview from field of tactile maps (and project) it would be suitable to mention (and comment) the project ATMAPS (www.atmaps.eu).

For more clarity of the references it would be suitable to format it as an alphabetical list.

Author Response

Please see the file attached. 

Reviewer 2 Report

In this paper the authors present a literature review that covers the current state of research in the field of algorithms and models for map generalization. The focus is on the existing solutions for automatic tactile map generation. The authors have looked at total number of over 500 papers published in this or similar areas. The reviewed papers cover in details several existing solutions in the field of automatic map production, as well as algorithms and models for data generalization that might be used for transformation of traditional spatial data into the haptic form, suitable for blind and visually impaired people. The conclusion however is that comprehensive solution for automatic tactile map generation does not exist.

There are many good things about this paper. The authors have chosen a interesting and not very well studied area. The number of papers covered in the study is quite impressive there are close to 100 references. As a result the related work section is quite detailed.

My major concern with the paper is that authors failed to discuss the specific issues of tactile map generation that makes this process different from the generic map generation. As a result most of the papers mentioned in the study seem unrelated to the main topic. Authors should discuss how this problem is different from the more general area fo map generalization and then discuss how the solutions in that area need to be modified in order to work for tactile maps.

Another issue with the paper is the awkward English. I strongly recommend the authors before the resubmit it again to have a native speaker proof read it. A small list of the typos that I found is listed below. Please address those as well.

Overall - as a study paper this one failed to discuss the topic in a necessary details. And with a lot of language issues this paper needs a lot more work.

Minor details

Page 3 synthesises -> synthesizes

Page 7 generalisation - > generalization

Page 2 fourier -> Fourier

Page 18 centres  -> centers

Page 18 analysing -> analyzing

Page 8 Istambul -> Istanbul

Page 8 compter -> computer

Page 8 Lonon -> London

Page 9 summarise -> summarize          

Author Response

Please see the file attached.

Reviewer 3 Report

This paper covers an interesting topic on automatic (tactile) map generation.

The research design for the literature review seems appropriate and it provide a good overview of the current status of this field. This is a valuable contribution.

However, I think that the paper can be strengthen in the discussion part that is missing an important part from my point of view.

Indeed, this is good to have the status of the field and what is the state-of-the art as well as the current limitations. But this would be good to also know from the authors what are the perspectives, what are the next 3 steps to move forward. Ideally, this can be used to draw a sort of research agenda in the field.

Therefore, I really encourage the authors to improve the discussion section with the above mentioned points.

Author Response

Please see the file attached.

Reviewer 4 Report

This is a very interesting bibliographic research that deals with a deep social issue of cartography, which obviously incorporates its practical dimensions in terms of facilitating the everyday life of people with disabilities.

The authors present excellent knowledge of the scientific content of the subject under consideration (maps and procedures in cartography). They deal with clear research questions and very well documented methodological steps to manage the sources in which they find their search data and finally, they proceed a scientifically documented way of assessing them in order to arrive at their conclusions.

In conclusion, I would say that this is one of the exceptional cases in the scientific literature about cartographic issues, where I personally cannot detect any flaws at any of the levels of assessment that a scientific article may receive.

That is why I suggest publishing the article as it is.

Author Response

Please see the file attached.
